# Microbiological Contamination of Urban Groundwater in the Brazilian Western Amazon

**Célia Ceolin Baia** [1,*], **Taíse Ferreira Vargas** [1], **Vivian Azevedo Ribeiro** [1], **Josilena de Jesus Laureano** [2], **Rachel Boyer** [3], **Caetano Chang Dórea** [3] **and Wanderley Rodrigues Bastos** [1,*]

1    Environmental Biogeochemistry Laboratory (WCP), Federal University of Rondônia (UNIR), Porto Velho 76801-058, Brazil
2    Limnology and Microbiology Laboratory, Federal University of Rondônia (UNIR), Ji-Paraná 76900-726, Brazil
3    Department of Civil Engineering, University of Victoria, Victoria, BC V8P 5C2, Canada
*    Correspondence: celia.ceolin@gmail.com (C.C.B.); bastoswr@unir.br (W.R.B.);
     Tel.: +55-69-99329-0257 (C.C.B.); +55-69-2182-2122 (W.R.B.)

**Abstract:** Groundwater is heavily exploited for a variety of uses. Depending on their structure, the wells from which water is extracted can act as an entry point/gateway for a variety of microbiological contaminants, which can cause numerous adverse health effects. This study aimed to identify the microorganisms present in the groundwater in the Western Amazonian city of Porto Velho, using a methodology that can be deployed in other city centers. We collected 74 water samples from both dug and drilled wells in March, August and November 2018. Total coliforms were detected in 96% of dug wells and 74% of drilled wells. Thermotolerant coliforms were found in 90% of dug wells and 61% of drilled wells. Biochemical identification indicated 15 genera of bacteria. The genera *Escherichia*, *Enterobacter*, *Cronobacter* and *Citrobacter* had the highest prevalence. The genera *Pseudomonas* and *Enterococcus* were also detected. Thermotolerant coliforms showed higher values when the water flow was higher. Our results indicate high fecal contamination and higher susceptibility to contaminants in shallow wells compared to deep wells. These findings reflect the precariousness of WASH (water, sanitation and hygiene) services and the importance of effective actions to combat groundwater degradation, improve the quality of the environment, and protect public health.

**Keywords:** pathogenic microorganisms; water quality; fecal coliforms; dug and drilled wells; Rondônia





## 1. Introduction

In 2015, the United Nations adopted a list of 17 Sustainable Development Goals (SDGs) to ensure equity and justice for all people. SDG 6 is to "ensure availability and sustainable management of water and sanitation for all" by 2030. However, it is estimated that currently, one in three people do not have access to safe drinking water [1]. Significant health consequences of consuming unsafe water include dysentery and other diarrheal diseases due to microbiological contaminants. The contamination of drinking water is steadily increasing in urban areas that are subject to demographic expansion, putting more and more people at risk [2].

One-third of the world's population uses groundwater as a drinking water source [3]. When available, groundwater is often considered to be a safer source than surface waters. Population growth through urbanization and anthropogenic activities such as industrial development and agricultural expansion can exacerbate groundwater quality issues by compromising the quality and amount of water in aquifers [4,5]. The discharge of untreated wastewater into surface water bodies and the construction of septic tanks without adequate safety criteria [6] has intensified groundwater pollution in many countries [7–9], especially in Latin America. This region has benefitted from increased levels of wastewater collection and onsite sanitation but still faces relatively low levels of adequately treated wastewater. In addition to the activities mentioned, the degree of groundwater contamination in well

water in this region can also be related to the depth and type of well, the operation and maintenance of the wells, and the presence of other point sources of pollution (e.g., landfills) [10,11]. These issues are often compounded by a lack of adequate safety criteria for drilling wells. Aquifer water quality also varies spatially and temporally, depending on the type of soil, rock formation and season [12,13].

In Brazil, groundwater plays an important role in drinking water supplies. The drinking water in 52% of cities is partially (16%) or fully (36%) supplied by groundwater [14]. Unfortunately, numerous studies have reported the presence of microbiological contaminants in groundwater globally [15–19], in Brazil [20,21] and in the Amazon region [22,23]. Braga et al. [24] for example, identified through cluster analysis the relation between the presence of *E. coli* and the rainy season in a region of the Maranhão Amazon. Contamination by total coliforms and *E. coli* were also found in groundwater that supplies schools in the state of Pará, Amazon region [24]. The authors point out that this can cause significant intestinal disorders in schoolchildren. In a city closer to our study area, Ji-Paraná (Rondônia), Ramos et al. [25] observed through principal component analysis that total coliforms had an important contribution to the formation of groundwater axes. Although this region has one of the highest rates of urban population growth, it remains understudied relative to other regions of the country—the most populous country on the continent.

The western Amazonian city of Porto Velho is a prime example of the recent increase in urbanization and the associated groundwater supply issues. In this study, a microbiological assessment of the groundwater supplying Porto Velho was conducted using a methodology that can also be deployed in other urban centers in the region. The assessment aimed to identify the species of bacteria present through biochemical techniques and to document possible associated factors such as well type, seasonality and water flow.

## 2. Materials and Methods

### 2.1. Study Area

The study was conducted in the city of Porto Velho (state of Rondônia) (Figure 1), located in the western part of the North Region of Brazil, within the Western Amazon (8°46′17″ S, 63°53′2″ W). During the last census (2010), the population of Porto Velho was 428,257, with 391,014 of those people living in urban areas/parts of the city [26]. The urban perimeter of the city is divided into 5 administrative zones (Law N° 840/1989), with the Center, Center South and Center North regions corresponding to Zone 1, the East region constitute Zones 2, 4 and 5 and the South region equivalent to Zone 3 [27]. The blue (34) and red (16) dots on the map represent the sampled wells (dug and drilled wells, respectively).

The state of Rondônia has a mainly tropical savanna climate with dry-winter characteristics (Aw, according to the Köppen classification). During the coldest months, the average temperature is approximately 18 °C. The dry season occurs during winter when rainfall rates are below 50 mm/month [28].

There are two aquifers within the urban perimeter of Porto Velho: the Içá Aquifer (free type) and Fraturado Norte Aquifer (semiconfined type). This latter aquifer supplies only 12.82% of the urban area. It covers a small area in Zones 1 and 3 and consists of rocks from the Santo Antonio Formation and a detrital–laterite cover, with low water potential due to its porosity and permeability (fissure type). The Içá Aquifer covers Zones 2, 4 and 5 and most of Zones 1 and 3. It consists of undifferentiated sedimentary cover, alluvial deposits and lake deposits. Due to its porous and permeable characteristics, it is a good water source [29]. The recharge areas are mainly located in the central part of the study area, covering areas of zones 1 and 2, where the potentiometric level is high and the flow is divergent. The discharge areas are in the topographic lows, where lower potentiometric values occur, covering areas of zones 2, 3, 4 and 5 [30]. As the city does not have adequate basic sanitation, the use of improperly constructed septic tanks and the release of effluents from industries and homes into the environment and the inappropriate disposal of garbage contributes to the degradation of groundwater quality.

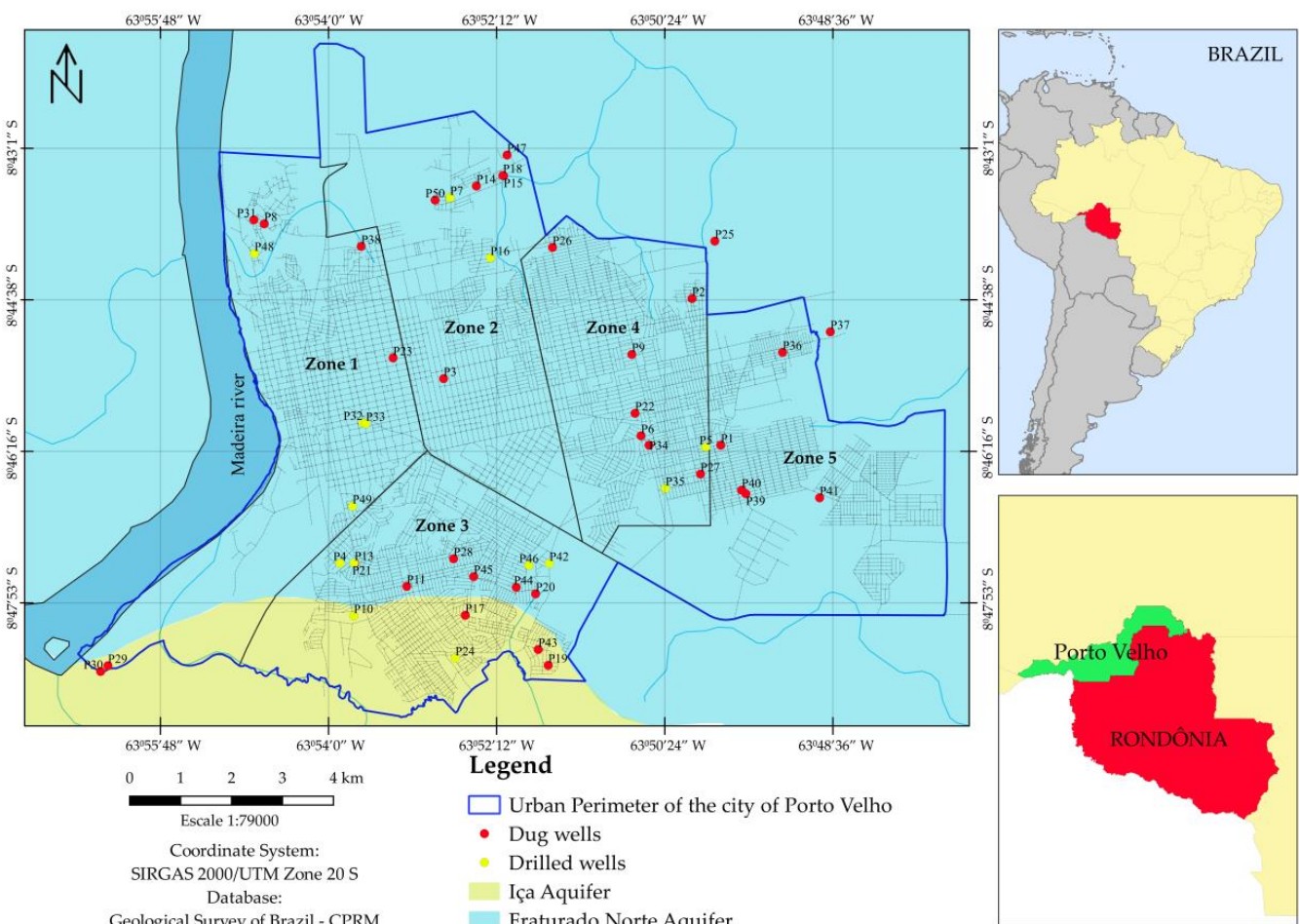

**Figure 1.** Urban area of the city of Porto Velho, showing the wells sampled.

Five lithostratigraphic units are distinguished in the study area: Undifferentiated sedimentary coverage, which makes up 85.2% of the area and are the most predominant in all zones. They are composed of alluvial and colluvial sediments formed by the deposit of sand, silt, clay or gravel. Alluvial deposits cover 2.3% of the area, with half of the area in zone 1 and small areas in the other zones, composed of sandy and clayey sediments considered to be unconsolidated to semi-consolidated. Lacustrine deposits, comprising 0.7% of the area in zones 2 and 3, is composed of clayey and silty sediments, massive or laminated and generally with a high presence of organic matter. Dendritic–lateritic covers, which correspond to 11.2% of the study area and cover small areas in all zones, are formed by ferruginized lateritic crusts and structures that may be composed of eluvial/coluvial debris. Santo Antônio Formation occupies 0.6% of Zone 1 and is made up of crystalline terrains of an igneous nature [31].

The wells that are most used by the residents of Porto Velho are dug wells, which capture water from aquifers closer to the surface. These wells can reach depths of up to 25 m. Drilled wells are used to obtain water from greater depths, down to 2000 m. In general, the 5 Zones have similar land use, with housing and commercial commerce. There are larger groups of commerce in Zone 1 and in Zones 4 and 5 are the most peripheral neighborhoods.

*2.2. Water Sampling*

This study was approved by the Research Ethics Committee of the Federal University of Rondônia (process 82891917.1.0000.5300). Water samples (a total of 74) were collected in March (34), August (16) and November (24) 2018 to represent the falling water, low water and rising water seasonal periods, respectively. Sampling locations were different

each month. The water was collected in 500 mL sterilized glass bottles from the nearest accessible outlet to the wells and kept under refrigeration (4 °C) until analysis.

### 2.3. Microbiological Analysis of Water Samples

A suite of microbiological indicators was used based on their global relevance in practice and the regulatory frameworks in place for the study area. *E. coli* is an important indicator of contamination by fecal coliforms and related pathogens [32]. In addition, coliform bacteria (total and thermotolerant), which include *E. coli*, can be used to evaluate the integrity and cleanliness of water distribution systems. Other bacteria from the *Enterobacteriaceae* family also have clinical importance because these microorganisms are frequently associated with gastrointestinal and intestinal diseases [33], and bacteremia and skin infections [34]. The National Environment Council of Brazil, through Resolution 396/2008, provides environmental guidelines for groundwater, including the acceptable concentration of enterococci. Resolution 275/2005 from the National Health Surveillance Agency specifies drinking water standards for mineral and natural water, based on several parameters including enterococci, *Pseudomonas aeruginosa* and *Clostridia* [35,36].

The water samples were analyzed within 24 h of collection. We used the membrane filter technique [37] for microbiological quantification. Volumes of 100 and 50 mL of each sample were filtered using a 47 mm diameter checkered cellulose membrane filter with 0.45 µm porosity (Millipore). The filters were placed in the following culture media: coliform agar (Merck KGA) and salmonella shigella agar (Himedia) for coliforms and other enterobacteria, cetrimide agar (Kasvi) for *P. aeruginosa*, and m-enterococcus agar (Acumedia) for enterococci. The dishes were incubated (Marconi MA-420 incubator) at 36 ± 2 °C for 24 h.

After the incubation period, pure colonies with morphological characteristics suggestive of enterobacteria were counted and selected. They were identified using the following biochemical tests: carbohydrate fermentation (triple sugar iron—TSI); hydrogen sulfide gas production and urease production (EPM), the use of citrate as an energy source, deamination of tryptophan, decarboxylation of the amino acid (MILI); ornithine decarboxylation, fermentation capacity of mannitol and bacterial motility [38,39].

To certify/ensure the adequate preparation of biochemical tests, we used the ATCC 13048 strain of *Enterobacter aerogenes* obtained from the Brazilian Research Center for Tropical Medicine (CEPEM, Rondônia) collection. For the bacterium *Pseudomonas aeruginosa*, the culture medium was specific, requiring no additional tests. For the genus *Enteroccocus*, only presumptive identification was performed

### 2.4. Data Analysis

The bacterial densities shown in Figures 2 and 3 were log10 transformed for better visualization. For *E. coli*, we used a non-detection limit value of 0.5 CFU (colony forming units), half of the method-detection limit, when we calculated the bacterial geometric means. To compare the two groups, we performed a Mann–Whitney U-test using GraphPad Prism 5. All statistical analyses had a significance level of $\alpha \leq 0.05$.

Groundwater flow was analyzed by plotting geographical coordinates and piezometric surface values. These data were used to generate a trend map of the direction of the groundwater flow with the Surfer 8 software package. The data were interpolated using the kriging method.

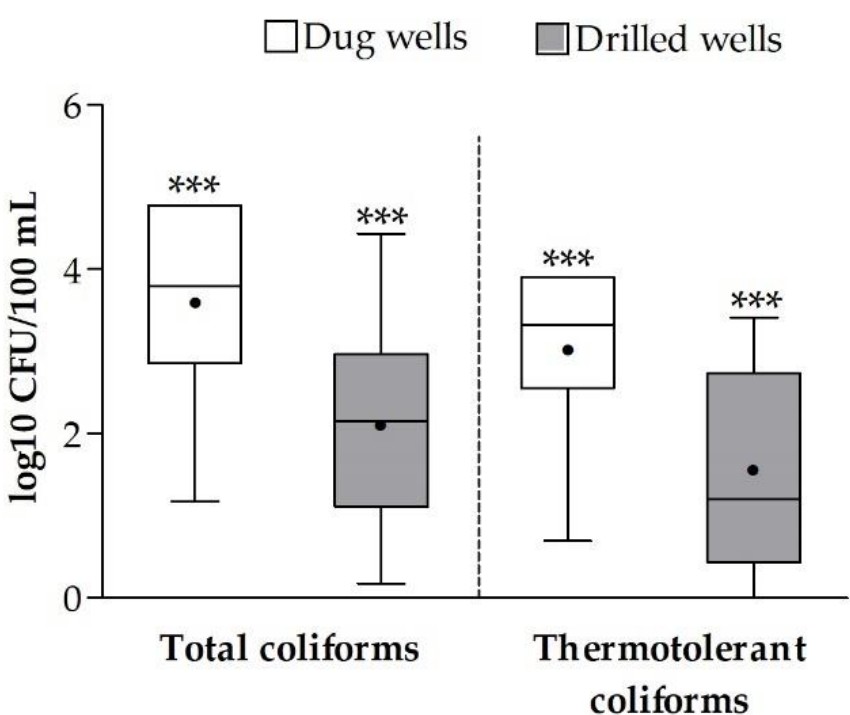

**Figure 2.** Box plot (min/max, lower/upper quartile, median, average) of CFU/100 mL of total and thermotolerant coliforms in the wells in Porto Velho (RO). Groups with significant differences are indicated by asterisks (***).

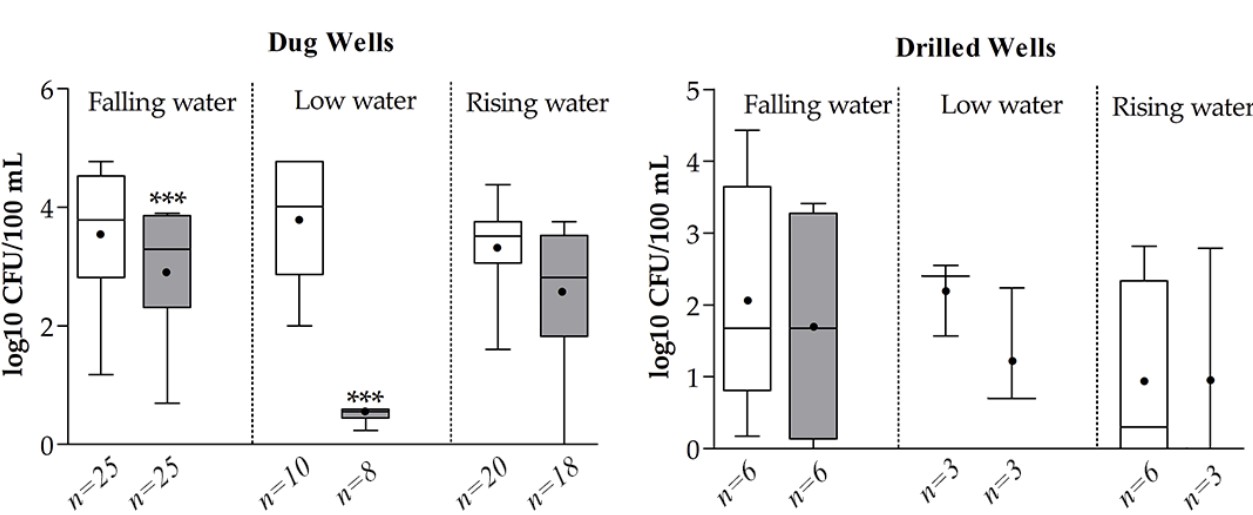

**Figure 3.** Box plot (min/max, lower/upper quartile, median, average) of the values of total and thermotolerant coliforms at the points sampled during different seasonal periods and from dug and drilled wells. Groups with significant differences are indicated by asterisks (***). *p*-values were 0.001 for dug wells and 0.5325 for drilled wells.

## 3. Results

### 3.1. Total and Thermotolerant Coliforms

Overall, total coliforms were detected in 89% of water samples, with values ranging between 1500 and 59,000 CFU/100mL. Thermotolerant coliforms were present in 81% of the water samples, with values ranging from 1 to 8000 CFU/100 mL. We also found that 96% of samples from dug wells had detectable levels of total coliforms and 90% had thermotolerant

coliforms. In the samples from the drilled wells, 74% contained total coliforms and 61% contained thermotolerant coliforms.

Figure 2 shows the density of total and thermotolerant coliforms in the contaminated wells. There were significant differences ($p < 0.0001$) between dug and drilled wells, with dug wells containing lower levels of both total and thermotolerant coliforms, indicating less contamination overall.

### 3.2. Seasonality

Figure 3 shows seasonal comparisons of microbial parameters during the three sampling periods, for both dug and drilled wells.

In the dug wells, the density of total coliforms was not significantly influenced by seasonality. Conversely, there was a statistical difference between the density of thermotolerant coliforms during the falling water and low water periods, with higher values during the falling water period. For the drilled wells, statistical tests did not indicate differences between the seasonal periods. These results may be due to the small sample size.

### 3.3. Biochemical Identification

Figure 4 shows which genera and species were identified in the groundwater and their respective percentages.

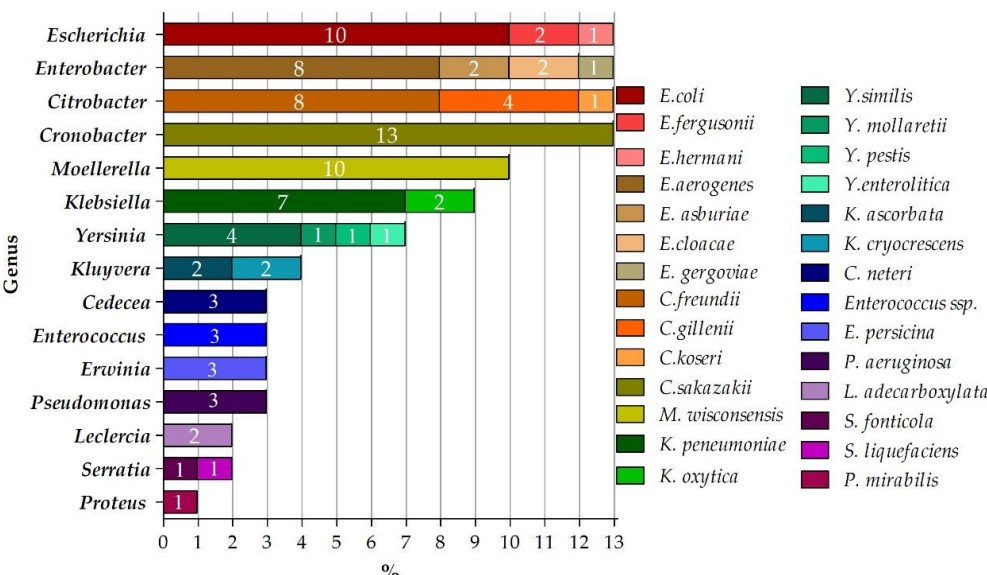

**Figure 4.** Distribution (%) of genera and species of bacteria in groundwater in Porto Velho, RO.

Fifteen genera were identified in the groundwater in Porto Velho. Four of the fifteen were dominant: *Escherichia*, *Enterobacter*, *Citrobacter* and *Cronobacter*. Each made up 13% of the total bacteria. These were followed by the genera *Moellerella*, *Klebsiella* and *Yersinia*, at 10%, 9% and 7%, respectively. The frequencies of all other genera were below 4%. In total, 27 species were identified. *Cronobacter sakasakii* was the most prevalent (13%), followed by *Escherichia coli* (10%), *Moellerella wisconsensis* (10%), *Enterobacter aerogenes* (8%), *Citrobacter koseri* (8%) and *Klebsiella pneumoniae* (7%).

### 3.4. Underground Flow

Figure 5 shows the spatial distribution of the density of thermotolerant coliforms and Figure 6 shows the direction of the underground water flow in the area where sampling occurred. Terrain topography was not a good predictor for water quality as, higher concentrations of thermotolerant coliforms were observed in Zones 2, 3 and 4 (higher elevations) and Zone 1 (lower elevations) contained the lowest density of thermotolerant coliforms.

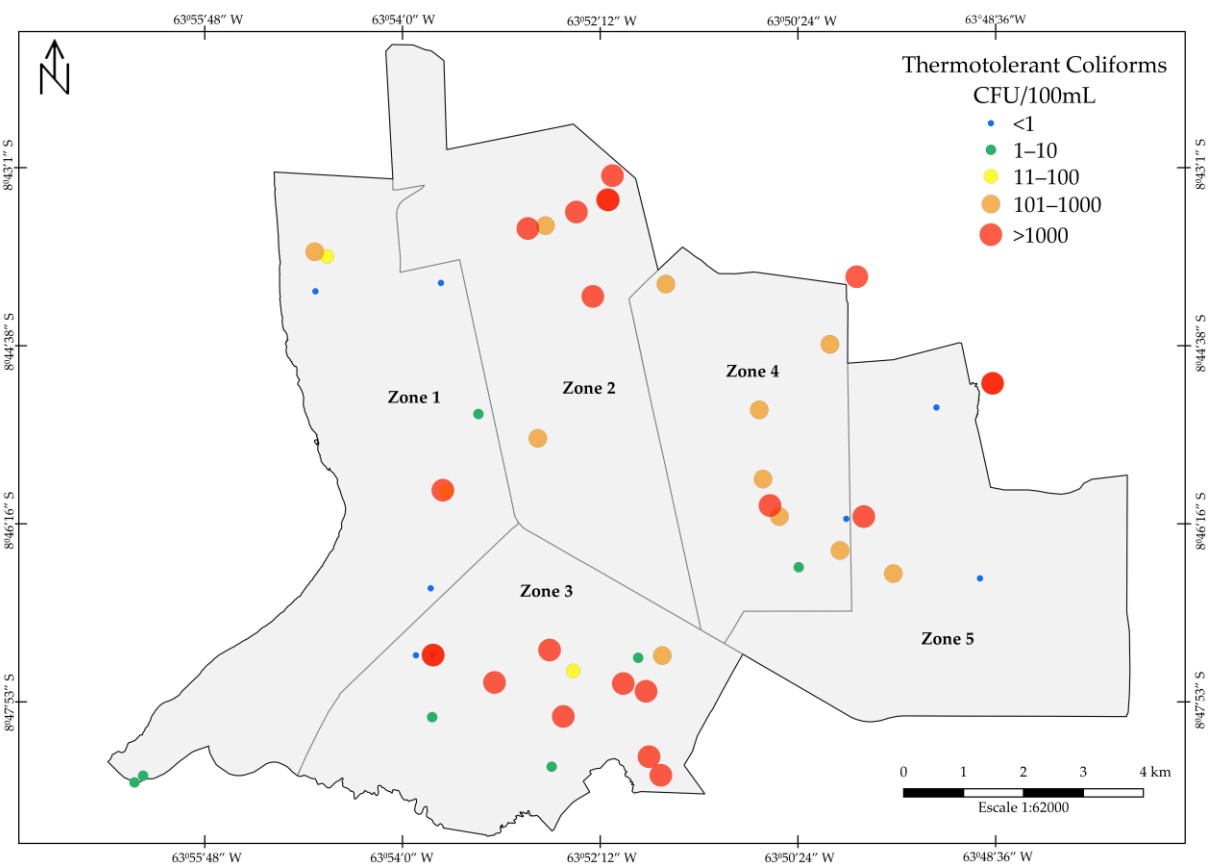

**Figure 5.** Spatial distribution of thermotolerant coliforms in groundwater in Porto Velho, RO.

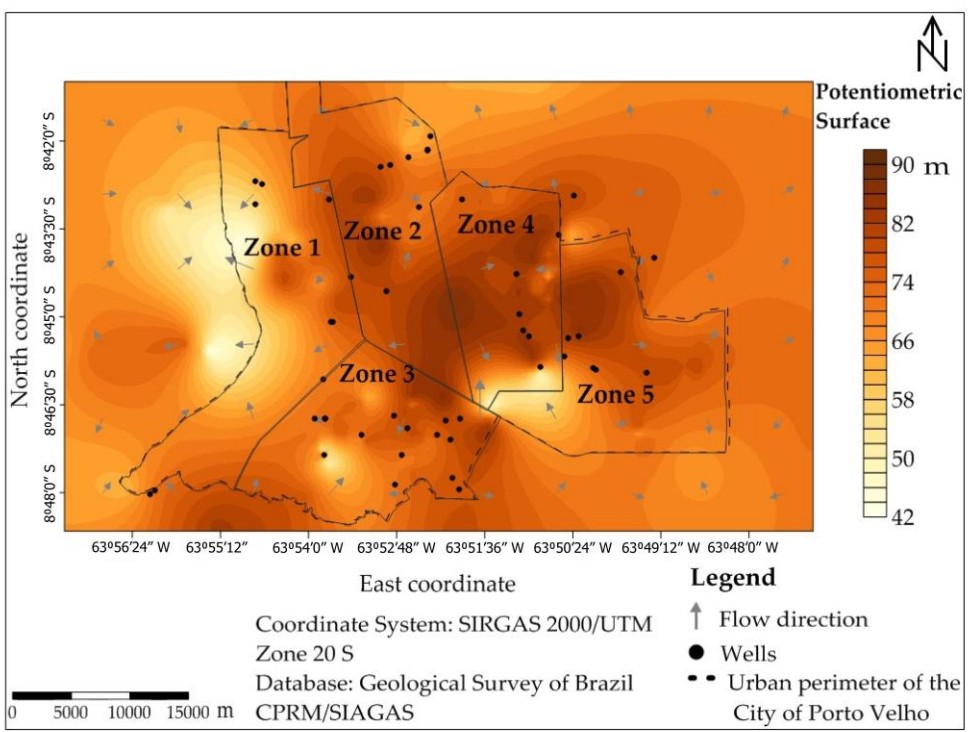

**Figure 6.** Underground flow within the urban perimeter of Porto Velho, RO.

## 4. Discussion

In this study, 81% of water samples (66 of 74) from wells (dug and drilled) showed the presence of high densities of thermotolerant coliform bacteria, surpassing the recommended limit set by the WHO [40] and Brazilian Legislation [41]. These findings suggest that there is a high level of environmental contamination and a large public health risk, as high densities of coliforms in the water can be indicative of the presence of other bacteria or pathogenic organisms such as *Giardia* and *Crysptoporidium* [40]. Further, the results reveal the impacts of inadequate water, sanitation and hygiene services, as well as the construction of septic tanks and wells that do not meet safety criteria and affecting wells and surface water bodies (streams and temporary ponds—igarapes) in the study area. These findings are also consistent with other studies in North Brazil [22,23].

The levels of thermotolerant coliforms in dug wells (shallower, with an average depth of 11.8 m) were significantly influenced by seasonality. Higher densities were detected in the falling water period (from May to July) than in the low water period (August to October). The greater flow during the other periods increased the contact of groundwater with sources of pollution. Differences in rainfall patterns due to climatic seasonality change watercourse patterns and affect water quality [42], the drilled wells showed no significant difference between the seasonal periods ($p = 0.5325$), likely because they are deeper (average depth of 36.2 m) and have less contact with pollution sources.

A total of 27 bacterial species were identified, with potentially pathogenic microorganisms exhibiting high variability in the groundwater samples. The genera *Escherichia*, *Enterobacter*, *Citrobacter*, *Serratia*, *Proteus*, *Klebsiella*, *Pseudomonas* and *Enterococcus* have also been reported in other groundwater surveys in Romania [15], Namibia [16] and Northeast and Southeast Brazil [21,43].

The most prevalent enterobacterium was *Cronobacter sakazakii* (13%), followed by *Escherichia coli* (10%) and *Moellerella wisconsensis* (10%). The species of the genus *Cronobacter* have been associated with pneumonia, conjunctivitis, diarrhea and urinary tract infections [44]. *C. sakazakii* mainly affects newborns, causing neonatal septicemia necrotizing enterocolitis [45,46]. Boamah et al. [47] also isolated species of *Cronobacter* and *Enterobacter* in shallow well water in Ghana.

The occurrence of *E. coli* indicates recent contact with fecal material. E. coli is normally found in the intestinal flora of humans and animals, where it is not a concern. However, it can lead to serious health impacts in other areas of the body, including urinary tract infections, bacteraemia, meningitis, and acute diarrhea [40]. The WHO cites safety measures to control potential risks of enteropathogenic *E. coli*, such as the protection of water sources from human and animal waste [40]. In our study area, we found that some wells were built according to suggested safety criteria, while abandoned wells were found without fences, exposing the aquifer to the external environment, and putting them at risk of fecal contamination.

The bacterium *Moellerella wisconsensis* may have negative health impacts. Its pathogenic power is not fully understood, but there is evidence of its relationship with diarrheal illness, with water being a likely source of infection [48,49]. Three other species that made up a high percentage of the sampled bacteria—Enterobacter aerogenes (8%), *Citrobacter freundii* (8%) and *Klebsiella peneumoniae* (7%)—have also been reported in groundwater in the state of São Paulo (Brazil) [21] and in India [19]. These enterobacteria can be found in the intestinal tracts of humans and animals, where they mainly cause infections in the respiratory and urinary tracts [50–52]. Two genera that do not belong to the enterobacterial family, *Pseudomonas* and *Enterococcus*, each had a prevalence of 3%. Only presumptive identification of *Enterococcus* was completed, and additional tests are required to determine the identification of the species level.

The presence of enterococci indicates fecal contamination, especially *E. faecalis*, an opportunistic bacterium that causes mostly urinary tract infections, endocarditis, and neonatal sepsis [53] and is in line with other drinking water studies that have isolated this bacterium [20,54,55]. A recent study in Northeastern Brazil showed enterococci values

in groundwater as high as 18 CFU/100mL, with contamination increasing during water storage and handling procedures [56].

The bacterium identified as *Pseudomonas aeruginosa* is an opportunistic and invasive species found in several environments. Its pathologies include serious nosocomial and human systemic infections, such as pneumonia, osteomyelitis, endocarditis, urinary and gastrointestinal infections, meningitis and septicemia, especially in immunocompromised people [57,58]. If water with a sufficient load of this organism is ingested or used for bathing by vulnerable people/populations (children, the elderly and hospital patients), it can cause infections on the skin and in the mucous membranes of the eyes, throat, nose and ears [40]. Hafiane et al. [17] and Bamigboye et al. [18] recently isolated *P. aeruginosa* in groundwater intended for human consumption and tested its resistance to antibiotics. Such studies highlighted the importance of assessing water quality for public health and for the relevant authorities to take action to provide safely managed drinking water services.

Water from regions with higher elevations flows to those with lower elevations due to gravitational force. If these higher regions are polluted, they can spread a plume of contamination to other lower-lying areas [59]. By investigating flow direction along with microbiological densities, we were able to identify specific points as potential microbiological contributors. For thermotolerant coliforms, two points with high potential and high densities of bacteria (P16 and P22 in Figure 6) may have influenced the density of contaminants at other nearby points (P26, P6 and P34 in Figure 6).

The high densities of microbiological contaminants reflect the large pollution load that comes mostly from septic pits that were built without following recommended safety standards as well as inadequate sewage treatment. Additionally, almost 90% of the study area has an unconfined aquifer [60], which is more vulnerable to contamination than confined aquifers because of greater permeability to flows containing various contaminants.

The results presented in this study point to the need to protect groundwater. CETESB—Environmental Company of the State of São Paulo, an important organization that acts in the preservation and recovery of water quality in Brazil, emphasizes the importance of some measures such as environmental licensing and inspection of potential sources of pollution, special projects to characterize aquifers, control of soil and mapping of vulnerability to the risk of groundwater pollution [61]. This last instrument is widely used and helps to identify areas of risk [62–64] and is a tool that decision-makers in the study area could use.

## 5. Conclusions

The microbiological data obtained in this study indicate the presence of a wide variety of genera from the group enterobacteria in the groundwater in Porto Velho. *Escherichia* (13%), *Enterobacter* (13%), *Citrobacter* (13%) and *Cronobacter* (13%), were the dominant genera. Dug wells were more vulnerable to contamination than drilled wells, (90% contamination in dug wells and 61% contamination in drilled wells), and the unconfined aquifer that is prevalent in the region facilitated contact with and transport of pollutant loads. The density of thermotolerant coliforms was higher in periods when water was in motion (falling water or rising water) compared to the low water period. Overall, our results highlight the importance of improving water, sanitation and hygiene services and adopting urgent preventive measures to improve the safety of groundwater in the city of Porto Velho.

**Author Contributions:** Conceptualization, C.C.B., C.C.D. and W.R.B.; methodology, C.C.B., T.F.V. and V.A.R.; software, C.C.B. and J.d.J.L.; validation, C.C.B. and T.F.V.; formal analysis, C.C.B., C.C.D. and W.R.B.; investigation, C.C.B., T.F.V., J.d.J.L. and V.A.R.; resources, C.C.B. and W.R.B.; data curation, C.C.B. and W.R.B.; writing—original draft preparation, C.C.B.; writing—review and editing, C.C.B., R.B., C.C.D. and W.R.B.; visualization, C.C.B., J.d.J.L. and R.B.; supervision, W.R.B.; project administration, W.R.B.; funding acquisition, W.R.B. All authors have read and agreed to the published version of the manuscript.

**Funding:** This research was funded by the Brazilian National Council for Scientific and Technological Research (CNPq, Grant no. 458977/2014-4 and 301912/2017-3).

**Data Availability Statement:** The datasets used and/or analyzed during the current study are available from the corresponding author upon reasonable request.

**Acknowledgments:** The authors gratefully acknowledge the staff of the Laboratory of Biogeochemistry of Federal University of Rondônia and all of the residents that allowed us to sample the water from their wells.

**Conflicts of Interest:** The authors declare no conflict of interest.

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
