# Peer review of "Microbiological Contamination of Urban Groundwater in the Brazilian Western Amazon"

_water, doi:10.3390/w14244023_

Round 1

Reviewer 1 Report

General comment

This study aimed to identify the microorganisms present in the groundwater in the Western Amazonian city of Porto Velho. Total coliforms in water samples collected from dug and drilled wells were detected and biochemical identification were carried out. The microbiological data obtained indicated the presence of a wide variety of genera from the group enterobacteria in the groundwater in Porto Velho. Dug wells were more vulnerable to contamination than drilled wells. The density of thermotolerant coliforms in high water period was higher than that in the low water period.

However, the present version is far from being ready for submission. Many supplements and modifications are required to improve the article quality. Just as important as the research contents, the language and grammar should be checked and revised carefully. Therefore, I recommend Major Revision. The author should revise the manuscript carefully according to specific comments below.

Specific Comments

1. Figure 1-

1) Number of all the sampling points should be displayed.

2) Fig.1 should be vectorized and drawn based on hydrogeological map.

2. Based on Figure 1, zoning basis of Zone 1-5 should be explained in Section 2.1. Moreover, geological and hydrogeological conditions and land utilization type of these five zones should also be added. 

3. In ‘Results’ section – the author should show the results of five zones respectively. Besides, differences and causes of groundwater quality among these zones should be analyzed accordingly.  

4. Figure 5-

1) Since the distribution of sampling points is inhomogeneous (Figure 1), spatial distribution of thermotolerant coliforms in groundwater should not be shown using auto-interpolation method. Try to use different symbol size to show different concentration ranges of thermotolerant coliforms instead.

2) Change the title ‘Spatial distribution of the levels of thermotolerant coliforms in the groundwater’ to ‘Spatial distribution of thermotolerant coliforms in groundwater’

5. Figure 6-

1) Higher quality of figures is required.

2) Use uniform font and font size in all the figures.

3) Some of the texts are overlapped.

6. In ‘Discussion’ section, the author should discuss the results of five zones respectively.

7. In ‘Conclusion’ Section, the author should draw the main conclusions of each zone accordingly.

Author Response

Comments on manuscript (ID: water-2050784) entitled ‘Microbiological contamination of urban groundwater in the Brazilian Western Amazon’

General comment

This study aimed to identify the microorganisms present in the groundwater in the Western Amazonian city of Porto Velho. Total coliforms in water samples collected from dug and drilled wells were detected and biochemical identification were carried out. The microbiological data obtained indicated the presence of a wide variety of genera from the group enterobacteria in the groundwater in Porto Velho. Dug wells were more vulnerable to contamination than drilled wells. The density of thermotolerant coliforms in high water period was higher than that in the low water period.

However, the present version is far from being ready for submission. Many supplements and modifications are required to improve the article quality. Just as important as the research contents, the language and grammar should be checked and revised carefully. Therefore, I recommend Major Revision. The author should revise the manuscript carefully according to specific comments below.

Specific Comments

  1. Figure 1-

1) Number of all the sampling points should be displayed.

Action: Sample point numbers have been entered in the figure.

2) Fig.1 should be vectorized and drawn based on hydrogeological map.

Action: The figure was vectorized and hydrological information was inserted.

  1. Based on Figure 1, zoning basis of Zone 1-5 should be explained in Section 2.1. Moreover, geological and hydrogeological conditions and land utilization type of these five zones should also be added.

Action: The city's zoning information was inserted, as well as more detailed information and specifications for the hydrological and geological area.

  1. In ‘Results’ section – the author should show the results of five zones respectively. Besides, differences and causes of groundwater quality among these zones should be analyzed accordingly.

Action: As per the previous item 2, in Section 2.1 of the manuscript we have further clarified that the division of the study area in 5 zones follows the administrative city divisions. There were no significant differences between these and we feel the use of such division will be of benefit to dissemination of our study’s findings to local authorities in charge of water and sanitation services. As such, we have kept our results as previously presented. We have applied a similar rationale to items 4 (Discussion) and 5 (Conclusion) of this rebuttal.

  1. Figure 5-

1) Since the distribution of sampling points is inhomogeneous (Figure 1), spatial distribution of thermotolerant coliforms in groundwater should not be shown using auto-interpolation method. Try to use different symbol size to show different concentration ranges of thermotolerant coliforms instead.

Action: Different sizes of symbols were used to show different concentration ranges of thermotolerant coliforms.

2) Change the title ‘Spatial distribution of the levels of thermotolerant coliforms in the groundwater’ to ‘Spatial distribution of thermotolerant coliforms in groundwater’

Action: The figure title has been changed.

  1. Figure 6-

1) Higher quality of figures is required.

Action: Figure adjusted for better quality.

2) Use uniform font and font size in all the figures.

Action: Fonts and font sizes have been standardized.

3) Some of the texts are overlapped.

Action: The figure has been adjusted but there are points that are too close to each other.

  1. Na seção 'Discussão', o autor deve discutir os resultados de cinco zonas, respectivamente.

Action: Adjusted.

  1. In ‘Conclusion’ Section, the author should draw the main conclusions of each zone accordingly.

Action: Adjusted.

"The manuscript has been proofread by professional services (Crayon-Bleu - https://www.crayon-bleu.com/) receipt available on request."

Reviewer 2 Report

This manuscript reported a study on microbiological assessment of groundwater quality in a local area of Brazil. The topic is of interest and importance. I think it can be considered for publication after some necessary improvements. My comments and suggestions are as follows.

1. In the second paragraph of Introduction, the authors mentioned that “Discharge of untreated wastewater into surface water bodies and the construction of septic tanks without adequate safety criteria has intensified groundwater pollution in many countries”. I think several necessary references should be cited here to support this statement. You mentioned many countries, so you need to cite some references from different countries. The following ones may be helpful: https://doi.org/10.1007/s12403-017-0258-6, https://doi.org/10.1007/s12403-016-0193-y, https://doi.org/10.1007/s12665-017-6787-2

2. I would also suggest the authors elaborate some research on groundwater in Brazil and the Amazon region. The authors cited relevant references, but they did not tell readers the research details. So please elaborate some of these studies in the third paragraph of introduction.

3. In the study area section, groundwater recharge and discharge should be introduced. In addition, groundwater development and activities that may affect groundwater quality should also be introduced in the study area section.

4. How many samples were collected in March, August, and November, respectively? The total number of samples is 74, but what about the number of samples in each month? Were the sampling sites the same in the three months? I guess only when the sampling sites are the same in the three months, can the comparison in this study be more meaningful.

5. The water samples were analyzed within 24 h of collection. However, the preservation measures for the samples were not introduced. Were they stored in 4 ℃?

6. I also suggest in this manuscript discussing the measures of groundwater quality protection.

7. I suggest add some quantitative results in the conclusion section.

8. References should be listed according to the journal style.

Author Response

This manuscript reported a study on microbiological assessment of groundwater quality in a local area of Brazil. The topic is of interest and importance. I think it can be considered for publication after some necessary improvements. My comments and suggestions are as follows.

  1. In the second paragraph of Introduction, the authors mentioned that “Discharge of untreated wastewater into surface water bodies and the construction of septic tanks without adequate safety criteria has intensified groundwater pollution in many countries”. I think several necessary references should be cited here to support this statement. You mentioned many countries, so you need to cite some references from different countries. The following ones may be helpful: https://doi.org/10.1007/s12403-017-0258-6, https://doi.org/10.1007/s12403-016-0193-y, https://doi.org/10.1007/s12665-017-6787-2

Action: Adjusted. The references mentioned have been implemented in the text.

  1. I would also suggest the authors elaborate some research on groundwater in Brazil and the Amazon region. The authors cited relevant references, but they did not tell readers the research details. So please elaborate some of these studies in the third paragraph of introduction.

Action: Some studies on the Amazon region were included in the text.

  1. In the study area section, groundwater recharge and discharge should be introduced. In addition, groundwater development and activities that may affect groundwater quality should also be introduced in the study area section.

Action: The required information has been inserted in the text.

  1. How many samples were collected in March, August, and November, respectively? The total number of samples is 74, but what about the number of samples in each month? Were the sampling sites the same in the three months? I guess only when the sampling sites are the same in the three months, can the comparison in this study be more meaningful.

Action: Adjusted. Sampling information has been inserted in the text.

  1. The water samples were analyzed within 24 h of collection. However, the preservation measures for the samples were not introduced. Were they stored in 4 â„ƒ?

Action: Adjusted. Preservation information has been inserted into the text.

  1. I also suggest in this manuscript discussing the measures of groundwater quality protection.

Action: A paragraph was inserted on the subject.

  1. I suggest add some quantitative results in the conclusion section.

Action: Some results have been inserted in the text.

  1. References should be listed according to the journal style.

Action: References have been revised.

"The manuscript has been proofread by professional services (Crayon-Bleu - https://www.crayon-bleu.com/) receipt available on request."

Reviewer 3 Report

Dear authors,

 The manuscript is an interesting study in the context of sustainable development, as it is necessary to ensure water for the entire population and to ensure sanitation for all accordingly to the Sustainable Development Goal SDG 6. Worldwide, groundwater quality is one of the major environmental problems and the efficiency of water quality control is mandatory since unsafe water consumption can cause serious health problems.

I have one minor comment:

Please improve the quality of Figure 6 because the point numbers are not visible;

 Thus, my decision is a MINOR REVISION.

Author Response

 The manuscript is an interesting study in the context of sustainable development, as it is necessary to ensure water for the entire population and to ensure sanitation for all accordingly to the Sustainable Development Goal SDG 6. Worldwide, groundwater quality is one of the major environmental problems and the efficiency of water quality control is mandatory since unsafe water consumption can cause serious health problems.

I have one minor comment:

Please improve the quality of Figure 6 because the point numbers are not visible;

 Thus, my decision is a MINOR REVISION.

Action: The figure has been adjusted. 

"The manuscript has been proofread by professional services (Crayon-Bleu - https://www.crayon-bleu.com/) receipt available on request."

Round 2

Reviewer 1 Report

The author have addressed many previous review comments but a few of issues remain.
Recommendation: minor revisions as per comments below.

1. Use uniform symbol of north arrow in Fig. 1, 5 and 6.

2. Letter 'N' in the north arrow in Fig. 1, 5 and 6 should be displayed in black. The current version is hard to read. 

3. Check initial letter case of five lithostratigraphic units (P3 L107 to 114 and L126 to 127).

Author Response

#Review 1

The author have addressed many previous review comments but a few of issues remain.
Recommendation: minor revisions as per comments below.

  1. Use uniform symbol of north arrow in Fig. 1, 5 and 6.

Action: The north arrow has been uniformed in the figures.

  1. Letter 'N' in the north arrow in Fig. 1, 5 and 6 should be displayed in black. The current version is hard to read.

Action: The letter 'N' was displayed in black.

  1. Check initial letter case of five lithostratigraphic units (P3 L107 to 114 and L126 to 127).

Action: Letters have been revised and corrected.

Reviewer 2 Report

My comments have been addressed, and I think the revised version can now be accepted for publication.

Author Response

#Review 02

My comments have been addressed, and I think the revised version can now be accepted for publication.

We are grateful for the reviewer's considerations.
